# Spatial and temporal variation in the occurrence of bottlenose dolphins in the Chesapeake Bay, USA, using citizen science sighting data

Lauren Kelly Rodriguez [1,2]*, Amber D. Fandel [1], Benjamin R. Colbert[1], Jamie C. Testa[1], Helen Bailey[1]

1 Chesapeake Biological Laboratory, University of Maryland Center for Environmental Science, Solomons, Maryland, United States of America, 2 Department of Integrative Biology, Michigan State University, East Lansing, Michigan, United States of America

* lrodriguez@umces.edu

**Data Availability Statement:** The data file with weekly dolphin sightings and weekly environmental data is available from the Dryad data repository (https://doi.org/10.5061/dryad.3tx95x6dg).

## Abstract

Bottlenose dolphins (*Tursiops truncatus*) are migratory marine mammals that live in both open-ocean and coastal habitats. Although widely studied, little is known about their occurrence patterns in the highly urbanized estuary of the Chesapeake Bay, USA. The goal of this study was to establish the spatial and temporal distribution of bottlenose dolphins throughout this large estuarine system and use statistical modeling techniques to determine how their distribution relates to environmental factors. Three years (April-October 2017–2019) of dolphin sighting reports from a citizen-science database, Chesapeake DolphinWatch, were analyzed. The dolphins had a distinct temporal pattern, most commonly sighted during summer months, peaking in July. This pattern of observed occurrence was confirmed with systematic, passive acoustic detections of dolphin echolocation clicks from local hydrophones. Using spatially-exclusive Generalized Additive Models (GAM), dolphin presence was found to be significantly correlated to spring tidal phase, warm water temperature (24–30°C), and salinities ranging from 6–22 PPT. We were also able to use these GAMs to predict dolphin occurrence in the Bay. These predictions were statistically correlated to the actual number of dolphin sighting reported to Chesapeake DolphinWatch during that time. These models for dolphin presence can be implemented as a predictive tool for species occurrence and inform management of this protected species within the Chesapeake Bay.

## Introduction

Bottlenose dolphins (*Tursiops truncatus*) are one of few apex pelagic predators within the Chesapeake Bay, USA. While the presence of these marine mammals has been previously reported [1, 2], their spatial distribution or relationship with environmental factors in this location has not yet been established. Dolphins are intelligent, wide-ranging cetaceans (marine

**Funding:** LKR was funded by a Maryland Sea Grant Research Experiences for Undergraduates fellowship. The Chesapeake DolphinWatch application was funded by the Chesapeake Bay Trust (grant #14.456) and Chesapeake Biological Laboratory. Thank you to the JES Avanti Foundation and Chesapeake Biological Laboratory for funding the hydrophones for the passive acoustic monitoring. We are also grateful for donations provided by members of the public to the project through the University System of Maryland Foundation that funded travel to deploy and recover the hydrophones. The funders had no role in study design, data collection and analysis, decision to publish, or preparation of the manuscript.

**Competing interests:** The authors have declared that no competing interests exist.

mammals) whose home ranges often overlap with areas of high anthropogenic activity. Because they are protected by the Marine Mammal Protection Act, the National Marine Fisheries Service is tasked with managing stocks (spatially-exclusive groups) of bottlenose dolphins. However, the lack of spatiotemporal data for these animals makes managing and protecting them in the Chesapeake Bay difficult.

The Chesapeake Bay is a large, highly urbanized estuary along the Mid-Atlantic coast of the USA, which is characterized by tourism, military activities, and shipping; all of which can disturb marine species [3]. The Bay is home to the two of the largest shipping ports in the USA (in Baltimore, Maryland and Hampton Roads, Virginia) as well as the largest naval base in the world (in Norfolk, Virginia). Dolphins that frequent this area are likely to encounter noise from recreational and commercial shipping as well as from naval training exercises. Understanding the spatiotemporal distribution of bottlenose dolphins within the Chesapeake Bay is an essential step to assessing what stressors local dolphins may be exposed to from these activities. This information would allow managers to consider potential impacts to this species in environmental assessments and therefore properly deconflict the presence of this protected species with anthropogenic events or require appropriate mitigation.

Although often residing along coasts, bottlenose dolphins can be very challenging and costly to survey [4]. However, information on wide-ranging, easily recognizable species can be obtained through the application of citizen science, the process of conducting research with the assistance of non-scientific volunteers [5, 6]. This method of data collection has been increasingly used to collect species presence data, especially with the advancement of mobile technology [7]. Opportunistic sightings by citizen scientists can greatly reduce the cost of data collection procedures, allowing investigation into the occurrence and phenology of critical species across broad spatiotemporal scales [8, 9].

In the Chesapeake Bay, the mobile and web-based public reporting application (app), Chesapeake DolphinWatch, has been used to collect opportunistic bottlenose dolphin sightings since 2017. This app allows citizen scientists to report sightings of dolphins, including the time, date, GPS location, number of animals observed, and pictures and video. These reported sightings provide a unique opportunity to study the distribution of bottlenose dolphins within the vast Chesapeake Bay and its tributaries, data that would be otherwise be highly resource-intensive to gather.

The distribution of dolphins in any area is likely affected by a number of both indirect and direct factors. Prior studies have shown that temperature, dissolved oxygen, and salinity influence the spatial and temporal distribution of bottlenose dolphins' favored ectothermic prey species along the Atlantic coast [10, 11]. Additionally, dolphins have shown behavioral changes in response to local variation in tidal phase, utilizing higher tidal ranges to efficiently hunt for prey near the coastline [12, 13]. These abiotic factors may be indirectly correlated with the occurrence of bottlenose dolphins.

Species Distribution Models (SDMs) are one of the tools applied by ecologists to identify underlying relationships between habitat characteristics (e.g., water quality, prey availability) and spatiotemporal patterns of animals, giving insight into their regional-specific habitat use [10, 14, 15]. These models may also be used to forecast both species occurrence and density, and are becoming increasingly popular in applications of dynamic ocean management [16–18].

In this study, we characterized the spatial and temporal distribution of bottlenose dolphins using sightings from the citizen science project, Chesapeake DolphinWatch. We utilized acoustic detections of dolphins at a site within the Bay to verify the observed temporal pattern of occurrence in the citizen science sighting reports. We then analyzed weekly dolphin sightings in relation to key abiotic factors to create SDMs for the Bay (divided into three segments).

The SDMs we developed provide a predictive tool that can be used to inform management and conservation efforts of this protected species within the Chesapeake Bay.

## Methods

### Study site and period

Residing along the northeastern coast of the USA, the Chesapeake Bay is an expansive estuary with an area of 11,600 km$^2$ [3]. Its mainstem extends through the states of Maryland and Virginia while its tributaries extend into regions of Washington D.C., Delaware, and Pennsylvania (Fig 1). To account for variability in environmental conditions throughout the estuary, and because the potential location error in the sightings reported by our citizen scientists was unknown, we spatially divided the Chesapeake Bay in this study into three latitudinally-equal segments (Upper, Middle, Lower) with the Lower segment being the one closest to the Atlantic Ocean (Fig 1).

Temporally, data (regarding species presence and environmental conditions) were collected from June 28, 2017 to October 14, 2019. Only data from April through October of each year were examined for this analysis. This does exclude the months of April, May, and June of 2017, in which data collection did not begin until late June.

### Dolphin sightings

Citizen scientists have been documenting bottlenose dolphin sightings on the Chesapeake DolphinWatch app since it launched on June 28, 2017. Initially, the Chesapeake DolphinWatch

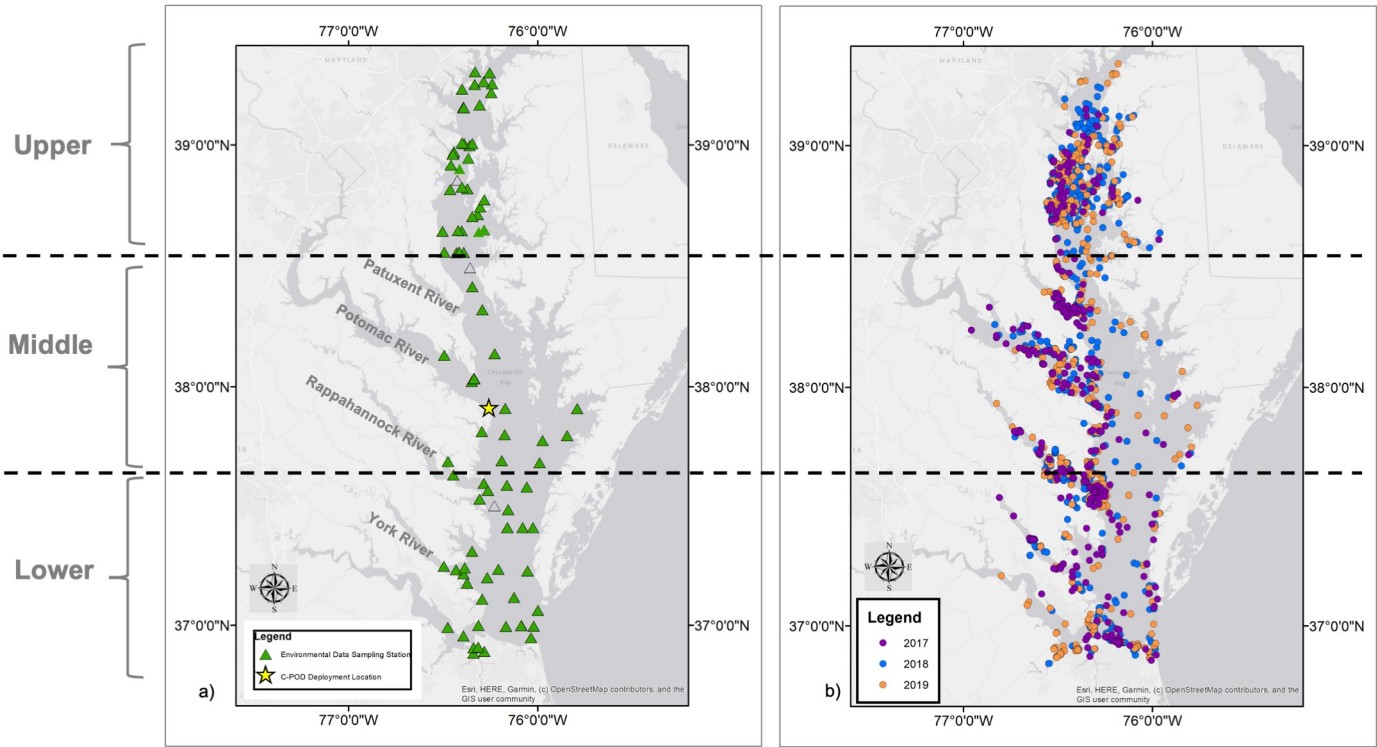

**Fig 1. Spatial distribution of study data.** a) The Chesapeake Bay study area, which we divided into three latitudinally-equal segments with the Lower Bay segment being closest to the Atlantic Ocean. The location of the C-POD passive acoustic monitoring device (yellow star) and locations of environmental data sampling stations (green triangles) are shown on the map. b) Spatial distribution of non-duplicated, confirmed bottlenose dolphin sighting reports from the Chesapeake DolphinWatch citizen science project. Map attribution: ESRI, HERE, Garmin, OpenStreet Map.

platform was web-based only (chesapeakedolphinwatch.org), but this was supplemented with a freely available mobile application for Android and Apple devices beginning in May 2018. From December 15, 2017 to April 1, 2018, the web-based platform was offline for maintenance and upgrades while the mobile application was developed. In this study, weekly dolphin sighting reports were analyzed from June 28, 2017 to October 14, 2019. Summing the observational reports per week reduced temporal sampling bias caused by higher reporting rates on weekend days.

Each sighting report entered into Chesapeake DolphinWatch required the location of the sighting, which could be obtained directly from the user's device or by the user selecting a point on a map. The time and date of the sighting were also recorded. A section for additional details on the sighting allowed users to include a description and/or pictures and videos of what they observed (Fig 2). Users could enter this information during the sighting or afterwards.

Application users were required to register and login with their email address so they could be contacted for further information about their sighting. For a reported sighting to be confirmed as a true sighting, it must meet the following requirements: 1) there is a photo or video of the encounter to accompany the sighting details provided and the location of the sighting is plotted in the water, or 2) there is a description of the sighting event. Some sightings from "trusted application users," a small group of people who were in contact with our team and regularly submitted reliable sighting reports, were confirmed even if no description or photographic evidence was included. Users periodically enter sightings with insufficient information to be considered a true sighting, and in these instances the data remain in the application but are deemed "unconfirmed". Sighting locations that did not match the entered description or that occurred on land were not confirmed. Unconfirmed sightings were not included in this analysis.

Duplicate reports of a dolphin group were excluded from our analyses. A mean of published bottlenose dolphin swim speeds indicated that their average swim speed is approximately 18.9 km/hour [19–21]. Based on this average speed, sighting reports were determined to be of the

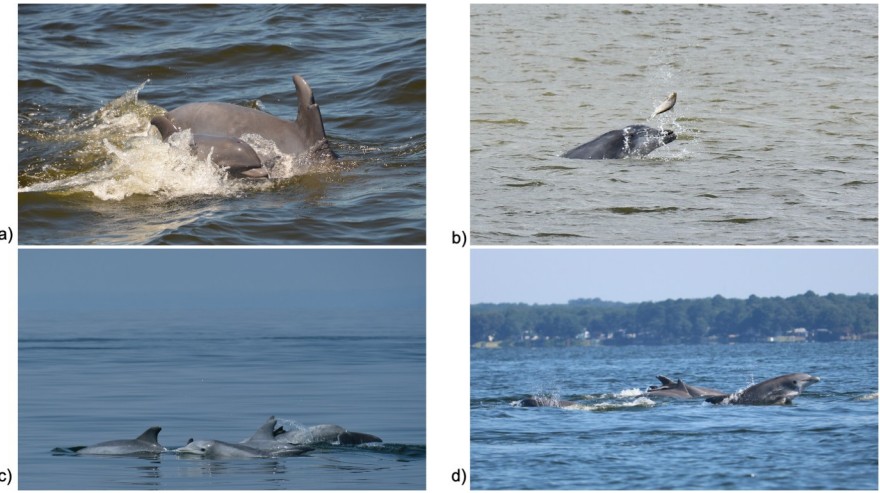

**Fig 2. Images submitted into the Chesapeake DolphinWatch application by citizen scientists.** a) Photo taken by Toni Knisley of dolphin pod swimming in the Patuxent River (38.30, 76.45) on July 8, 2018. b) Photo taken by Kim Chase Brown of dolphin foraging in the Lower Bay (36.84, -76.55) on October 8, 2018. c) Photo taken by Rhiana Scholz of dolphins swimming in the Kent Narrows channel within the mainstem part of the Bay (38.83, -76.40) on July 3, 2018. d) Photo taken by Tania Richardson Remaly of dolphin pod near Ragged Island (38.15, -76.57) on July 9, 2017.

same pod if the sighting occurred within 9.5 km of each other during a 30-minute time period. Only confirmed, nonduplicated sightings of bottlenose dolphins were included in the final dataset used for these analyses. The temporal occurrence and distributional patterns of bottlenose dolphin sighting reports across the Chesapeake Bay were then examined using ArcMap 10.6 (Fig 1B).

## Acoustic dolphin detection

The Chesapeake DolphinWatch citizen science project provided a network of observers over a large spatial extent of the Bay, but its unsystematic nature may have affected the temporal patterns observed in sightings. People are generally more active on the water during summer months in the study area, which creates temporal bias in citizen science reports. To minimize this temporal bias, dolphin sighting reports were analyzed weekly from June 2017 to October 2019. We also used acoustic detections of bottlenose dolphins to verify the seasonal occurrence pattern observed.

A C-POD (Chelonia Wildlife Acoustic Monitoring, www.chelonia.co.uk) was deployed in the middle Bay at the mouth of the Potomac River (37.914˚N, 76.258˚W) from May through October during each study year (Fig 1A). No acoustic data for April, the first month of our study period, was available. C-PODs are automated cetacean click detectors that record the time, date, intensity, duration, and frequency of dolphin click events. Because this site was relatively shallow (3 meters), the high and moderate quality clicks trains from the C-POD's KERNO classifier were manually verified. The number of hours with dolphin detections (detection positive hours, DPH) was summed daily to account for the conservative nature of C-POD detections [22]. Daily DPH were summed weekly (mean DPH per week) and compared to the seasonal pattern of dolphin sightings reports from Chesapeake DolphinWatch in the Middle Bay (Fig 3C). To determine whether the acoustic detections and reported sightings showed similar trends in dolphin presence, a Spearman's Rank Correlation test in R (version 4.0) using the cor.test function was utilized [23].

## Environmental data

Environmental data from June 28, 2017 through October 14, 2019 were obtained using the same temporal (weekly) and spatial (Lower, Middle, Upper Bay segments) resolution as the divisions of the Chesapeake DolphinWatch sightings data. These environmental data included tidal phase (spring/neap), temperature (in degrees Celsius), salinity (in parts per thousand, PPT), and dissolved oxygen (in milligrams per liter, mg/L). Because dolphins were not sighted in the upper parts of tributaries, we excluded environmental data from these segments on both the western and eastern coasts. Environmental data from 76 sites were obtained from the open-source databases of Maryland's Department of Natural Resources (http://eyesonthebay. dnr.maryland.gov), the National Oceanic and Atmospheric Administration (https://buoybay. noaa.gov), and the Chesapeake Bay Program (http://data.chesapeakebay.net/WaterQuality).

The fraction of the moon illuminated per night was obtained from the Astronomical Applications Department of the US Naval Observatory to serve as a proxy for tidal phase (http://aa. usno.navy.mil/index.php). These data were preprocessed into weekly average values (ranging on a scale from 0–1.0) and categorized as either "spring" or "neap" tides depending on the lunar phase. Spring tides occur at every new moon (0) or full moon (1.0), whereas neap tides which occur during intermittent lunar phases [13]. Fractional moon illumination greater than 0.7 (full moon) or less than 0.3 (new moon) were denoted as spring tides and a moon illumination between, 0.4–0.6, was classified as a neap tide.

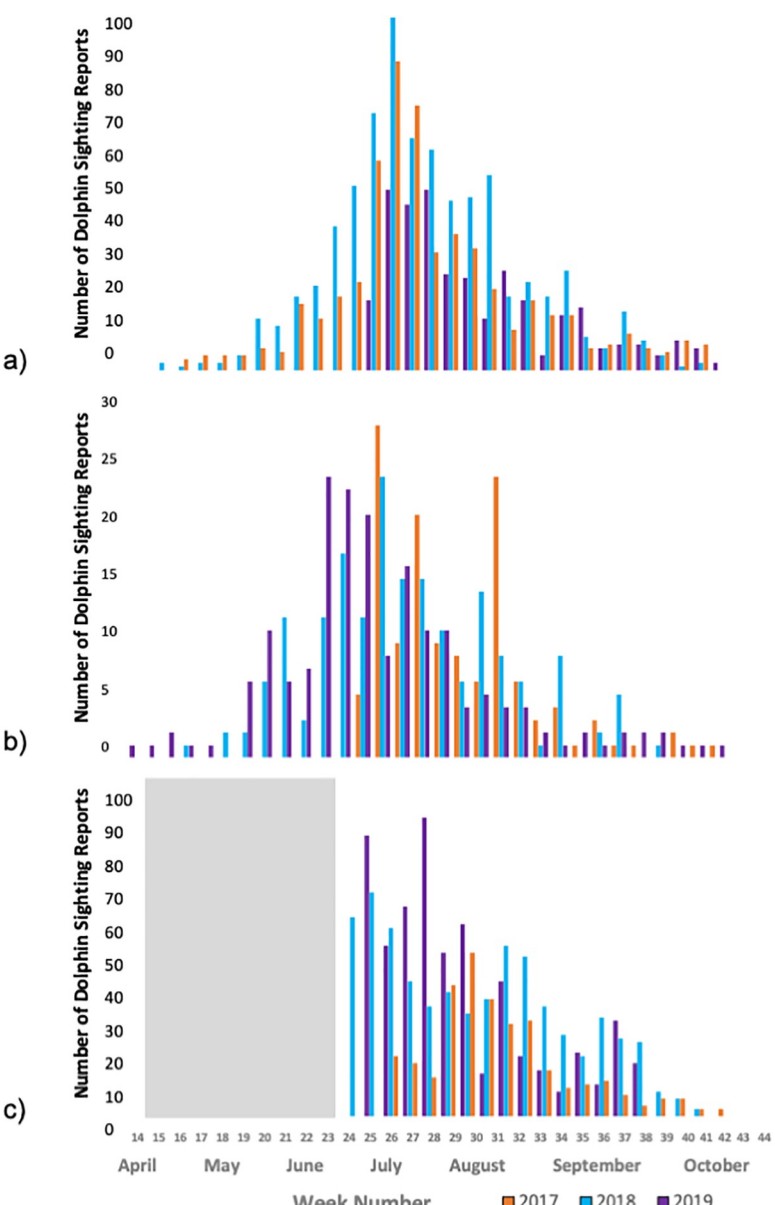

**Fig 3. Temporal distribution of dolphin data.** a) Weekly total of confirmed, non-duplicated Chesapeake DolphinWatch sighting reports across all segments of the Chesapeake Bay. b) Weekly total of confirmed Chesapeake DolphinWatch sighting reports in the Middle Bay segment of the Chesapeake Bay. c) Total sum of weekly acoustic detection positive hours (DPH) for the Middle Bay segment only, with the gray area indicating the absence of data. Map attribution: ESRI, HERE, Garmin, OpenStreet Map.

## Statistical analysis

Collinearity, instances in which two or more explanatory variables were correlated, was determined by examining the variance inflation factor (VIF). The VIF, calculated in R [24], detects collinearity among explanatory variables using a regression analysis. VIF was calculated prior to any model construction—any variable with a VIF greater than 3 was removed from further analysis [25].

The relationship between weekly dolphin sightings and environmental conditions (tidal phase, temperature, salinity, and dissolved oxygen) was investigated using generalized additive models (GAM) with a Poisson error distribution and log link function. GAMs serve as an SDM, accounting for nonlinear relationships between the response variable (weekly dolphin occurrence) and dependent predictor variables (environmental conditions) [26].

Tidal phase was included as a categorical explanatory variable, coded as either "spring" (1) or "neap" tide (0). Temperature, salinity, and dissolved oxygen were continuous numerical explanatory variables. The total number of Chesapeake DolphinWatch application users per week was included as an explanatory variable to account for the seasonal variation in app user frequency. Application user data was obtained from Chesapeake DolphinWatch's Google Analytics reports (S1 Fig). One GAM was conducted for each the Lower, Middle, and Upper segments of the Chesapeake Bay using the "mgcv" package in R [27]. Smoothing functions (used on each variable) were limited to four degrees of freedom for each predictor variable to avoid overfitting the data.

Concurvity measures the similarity of two or more predictor variables' relationships with the response variable in a model [28]. This parameter was calculated after construction of each model using the "mgcv" package [27]. If a variable exceeded a concurvity of 0.8 (on a scale from 0–1.0), it was deemed significantly correlated with another predictor variable and removed from the mode [25]. Variables which fit the expected values for both collinearity and concurvity were kept in the final models.

Each GAM was created using data from June 28, 2017 through June 1, 2019. The remaining data from 2019 (June–October) was used as a testing dataset to evaluate the ability of the model to predict dolphin occurrence. The function predict.gam [27] was utilized, which predicted dolphin occurrence given the observed values for environmental data and Google Analytics user data. The predictions were compared with the observed number of weekly sightings reported by Chesapeake DolphinWatch users during that time period using a two-sided Spearman's rank-order correlation test. The Root Mean Square Error (RMSE) was also calculated for each Bay segment model's predictions to determine which predictions were most accurate.

## Results

### Dolphin sightings

A total of 2,907 reported sightings were obtained through the Chesapeake DolphinWatch between June 2017 and October 2019. These sightings were reported by 953 registered users, 14 of which were deemed "trusted application users" (approximately 2% of users). A total of 1,788 individual bottlenose dolphin sighting reports were confirmed and included in the final analyses (68% of all reported sightings).

There was interannual variation in bottlenose dolphin sightings (Fig 3A). More sightings occurred in all three segments of the Bay in 2018 (Table 1), though each year showed a similar intraseasonal pattern of sightings. In each year, the peak number of sightings occurred during

**Table 1. Chesapeake DolphinWatch database summary statistics.**

| Year | Total Sighting Reports | Confirmed Sighting Reports | Percentage of Confirmed Sightings |
|------|------------------------|----------------------------|-----------------------------------|
| 2017 | 931 | 419 | 45.00 |
| 2018 | 1135 | 822 | 72.42 |
| 2019 | 841 | 547 | 65.04 |

Summary information for yearly dolphin sighting reports submitted into the Chesapeake DolphinWatch application.

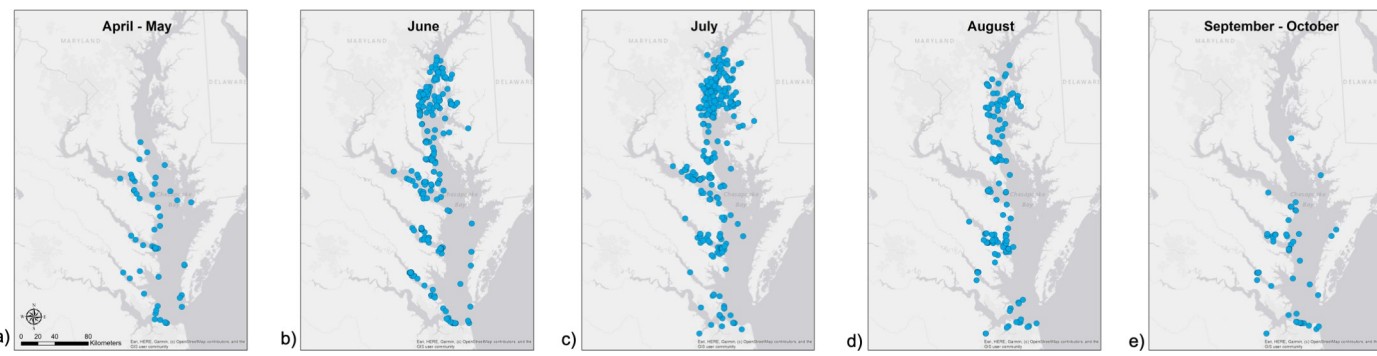

**Fig 4. Spatial distribution of confirmed, non-replicated bottlenose dolphin sighting reports from Chesapeake DolphinWatch for 2018.**

the first week of July (week 27; Fig 3A). In 2017, 49 confirmed sightings were reported during this week; in 2018, 96 confirmed sightings were reported; and in 2019, 84 confirmed sightings were reported. The week with the fewest sighting reports was also the same in each year–the final week of October (week 44; Fig 3A). We focused our study analysis on data from April to October of each year as the application was offline during winter 2017–2018 and sighting reports were very low during winter 2018–2019, which could have resulted from less outdoor activity by our citizen science network.

The spatial distribution of bottlenose dolphins throughout the Chesapeake Bay showed a seasonal pattern. Dolphins were first reported (first week of April, week 14) in the Lower Bay segment (Fig 4). During summer months (June-August), they were reported in all three segments of the Bay. The final sighting reports of the years occurred only in the Middle and Lower Bay segments at the end of October (week 44). Sightings in September and October were primarily in the Lower and southern portion of the Middle Bay (Fig 4).

Sighting reports were primarily concentrated around the coastal areas of the Chesapeake Bay, although they did also occur in the mainstem. Dolphins were reported at the mouths of multiple tributaries, primarily in the Potomac River, Rappahannock River, and the York River, though these sightings did not extend far into the rivers (Fig 5). The highest frequency (n = 136) of sightings was located at the mouth of the Rappahannock River (37.57˚N, -76.34˚W) in the Middle Bay. The majority (74%) of these reports were submitted to the app by a single observer. Note that we were unable to test if the Rappahannock River was truly the most significant location for dolphin sightings or if this user's diligent efforts created spatial bias. Many sightings also occurred in the western part of the Upper Bay and the central part of the Lower Bay. Though sighting densities varied spatially, the sightings reported across the study period within each segment were similar (ANOVA: $F_{2,231}$ = 1.629, p = 0.198). The Upper Bay had 664 confirmed sighting reports over three seasons, the Middle Bay had 556 sightings, and the Lower Bay had 650 sightings.

## Acoustic dolphin detections

Detection positive hours (DPH) from the C-POD (Fig 3C) followed a seasonal pattern similar to the sighting data in the Middle Bay (Fig 3B). Each year, the total DPH peaked in July and August, with the highest number recorded in July 2017 (Fig 3C). There was a significant correlation between the frequency of weekly acoustic detections and weekly sightings from 2017 through 2019 (Spearman's rank correlation: rho = 0.73, p < 0.01).

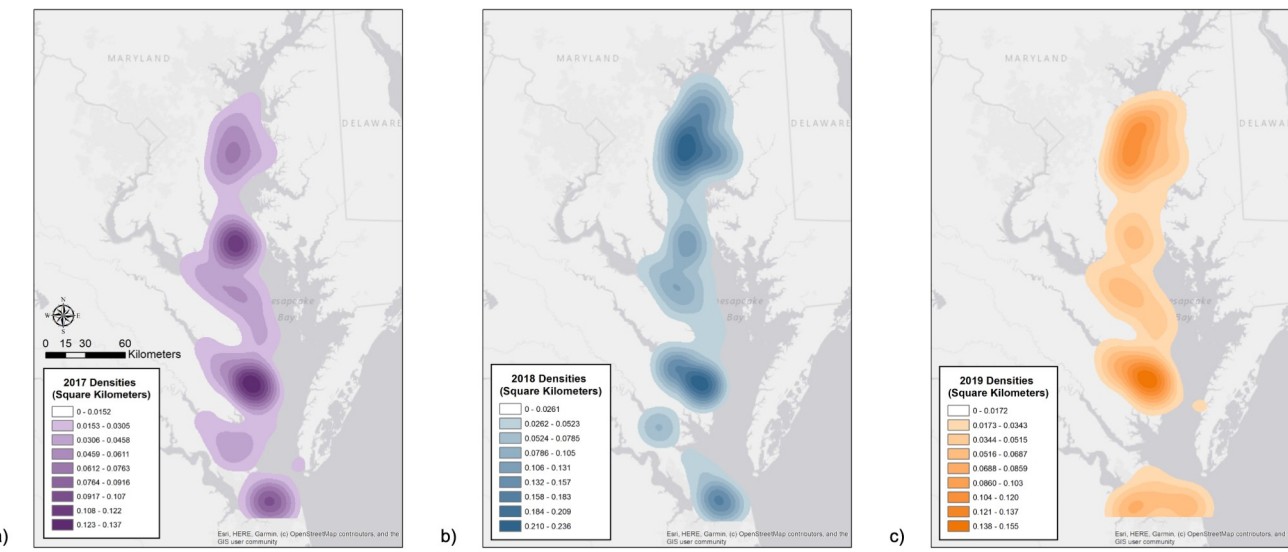

**Fig 5. Spatial density of dolphin sightings.** Maps showing the densities (sightings per kilometers$^2$) of confirmed bottlenose dolphin sighting reports from Chesapeake DolphinWatch (a– 2017, b– 2018, c– 2019). Darker colors indicate higher densities. Map attribution: ESRI, HERE, Garmin, OpenStreet Map.

## Statistical analysis

All continuous predictor variables–temperature, salinity, and dissolved oxygen—had variance inflation factors (VIFs) below 3. Therefore, these variables were retained in the models. The weekly sightings of bottlenose dolphins were significantly related to temperature and salinity in all three segments of the Bay (Table 2). Variations in temperature and salinity explained greater than 50% of the variance in dolphin occurrence in all GAMs. Dolphins were sighted more often when water temperatures in the Chesapeake Bay was 20˚C or higher (Fig 6). The relationship between weekly bottlenose dolphin sightings and salinity varied with respect to latitude (Table 2, Fig 6). In the Lower Bay, bottlenose dolphins were most often sighted when the salinity was 16 PPT or higher. In the Middle Bay, dolphins were sighted most frequently in salinities ranging from 6–18 PPT. In the Upper Bay, maximum dolphin sightings occurred in lower saline conditions (5–10 PPT).

Tidal phase was significantly related to bottlenose dolphin sightings throughout the Lower and Middle Bay. Dolphins were sighted more often during spring tides compared to neap tides. In the Upper Bay, the tidal phase was not significantly related to the number of dolphin sightings.

Initial concurvity estimates indicated that the response of bottlenose dolphins to marine dissolved oxygen was significantly correlated to their response to both temperature and salinity in the Upper Bay segment (Table 3). Upon re-running each model excluding dissolved oxygen included as an explanatory variable, the variance explained parameter decreased minimally. To mitigate the confounding relationship among the variables and maintain compatibility between models, dissolved oxygen was excluded from the final GAMs for each Bay segment.

Dolphin occurrence predictions (using the final GAM models) were significantly correlated with the observed dolphin sightings between June and October 2019 in the Middle and Upper Bay (Lower Bay: p = 0.11, Middle Bay: p< 0.01, Upper Bay: p< 0.01, Fig 7). The GAM for the Middle Bay yielded the most accurate predictions of weekly dolphin occurrence (RMSE = 2.84).

**Table 2. Results from the generalized additive model (GAM).**

| Lower Bay | | | | |
|---|---|---|---|---|
| **Parametric Coefficients** | | | | |
| | Estimate | Standard Error | z value | Pr($<$\|z\|) |
| Intercept | -3.91 | 0.09 | -42.71 | $< 0.01^*$ |
| Spring Tides | 0.20 | 0.09 | 2.06 | $0.04^*$ |
| **Smoothing Terms** | | | | |
| | Estimated degrees of freedom | Reference degrees of freedom | Chi-squared | P-value |
| Temperature (˚C) | 2.59 | 2.86 | 108.09 | $< 0.01^*$ |
| Salinity (PPT) | 2.06 | 2.43 | 70.32 | $< 0.01^*$ |
| $R^2$: 0.04, Deviance Explained: 52.6% | | | | |
| **Middle Bay** | | | | |
| **Parametric Coefficients** | | | | |
| | Estimate | Standard Error | z value | Pr($<$\|z\|) |
| Intercept | -4.63 | 0.11 | -41.63 | $< 0.01^*$ |
| Spring Tides | 0.49 | 0.11 | 4.39 | $< 0.01^*$ |
| **Smoothing Terms** | | | | |
| | Estimated degrees of freedom | Reference degrees of freedom | Chi-squared | P-value |
| Temperature (˚C) | 1.00 | 1.00 | 137.10 | $< 0.01^*$ |
| Salinity (PPT) | 2.16 | 2.59 | 7.46 | $0.05^*$ |
| $R^2$: 0.33, Deviance Explained: 52.5% | | | | |
| **Upper Bay** | | | | |
| **Parametric Coefficients** | | | | |
| | Estimate | Standard Error | z value | Pr($<$\|z\|) |
| Intercept | -4.85 | 0.17 | -29.20 | $< 0.01^*$ |
| Spring Tides | 0.05 | 0.10 | 0.51 | 0.61 |
| **Smoothing Terms** | | | | |
| | Estimated degrees of freedom | Reference degrees of freedom | Chi-squared | P-value |
| Temperature (˚C) | 2.96 | 3.00 | 92.19 | $< 0.01^*$ |
| Salinity (PPT) | 2.51 | 2.84 | 62.67 | $< 0.01^*$ |
| $R^2$: 0.86, Deviance Explained: 76.8% | | | | |

Each model compares average weekly bottlenose dolphin occurrence in response to average weekly tidal phase (spring or neap), temperature (˚C), and salinity (parts per thousand, PPT).

## Discussion

Bottlenose dolphins have not previously been considered to be regular visitors to the Chesapeake Bay. This study describes the seasonal presence of dolphins in this area and relates their occurrence to temperature and salinity, two environmental parameters that are continuously and systematically measured. As an urbanized, coastal region, it's crucial for species management agencies in and around the Chesapeake Bay to assess the spatiotemporal occurrence of protected species, such as bottlenose dolphins. The peak in dolphin sightings during the summer season indicates that managers will need to take appropriate measures to alleviate anthropogenic activity that may interfere with bottlenose dolphin acoustic communication and foraging techniques during this period. Between regional naval operations, urban coastal construction, and recreational activity, the Chesapeake Bay is a relatively loud and crowded marine environment.

Previous studies on Atlantic bottlenose dolphins describe their distribution as being affected by the density of prey species [13, 29, 30]. Although information is scarce on

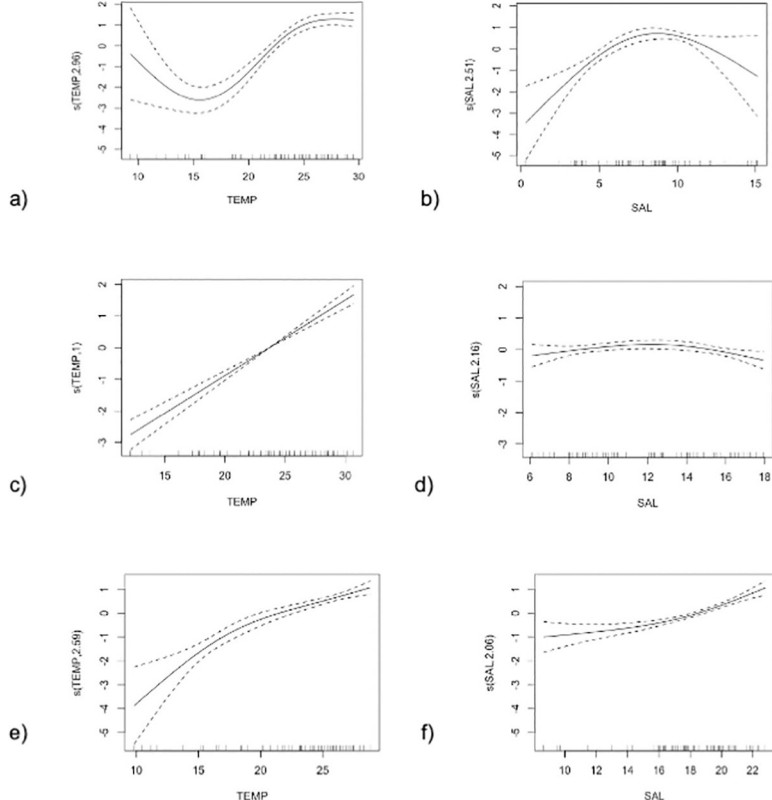

**Fig 6. GAM smoothers.** The relationship between weekly occurrence of bottlenose dolphins and water temperature (˚C; a, c, e) and salinity (part per thousand, PPT; b, d, f) for each Bay segment. Panel a and b show results from the Upper Bay, c and d show results from the Middle Bay, and e and f show results from the Lower Bay. Explanatory variables are on the x-axes with tick marks showing the distribution of underlying data and the centered, fitted values are on the y-axes. Confidence intervals are shown as dashed lines.

bottlenose dolphin diet in the Chesapeake Bay, a review of diets from bottlenose dolphins near North Carolina, Virginia, and Maryland indicated that they most frequently fed on weakfish (*Cynoscion regalis*), Atlantic croaker (*Micropogonias undulates*), and spot (*Leiostomus xanthurus*) [11]. Survey data for these prey species were not available for our study period. Instead, we utilized environmental parameters as a proxy for factors that influence the lifestyles of these ectothermic prey.

Water temperature, a significant explanatory variable in our models, influences juvenile development within these species [31]. During summer months, the Chesapeake Bay reaches temperatures that are high (27˚C) relative to offshore marine waters (20˚C), providing nursery habitat for hundreds of species of fish, including dolphins' prey [32, 33]. Weakfish, croaker,

**Table 3. Concurvity estimates for all predictor variables.**

| Bay Segment | Temperature Concurvity Estimate | Salinity Concurvity Estimate | Dissolved Oxygen Concurvity Estimate |
|---|---|---|---|
| Upper | 0.799 | 0.219 | 0.834* |
| Middle | 0.321 | 0.083 | 0.170 |
| Lower | 0.497 | 0.112 | 0.051 |

Estimates greater than 0.8 (marked with an asterisk) were removed from the final model.

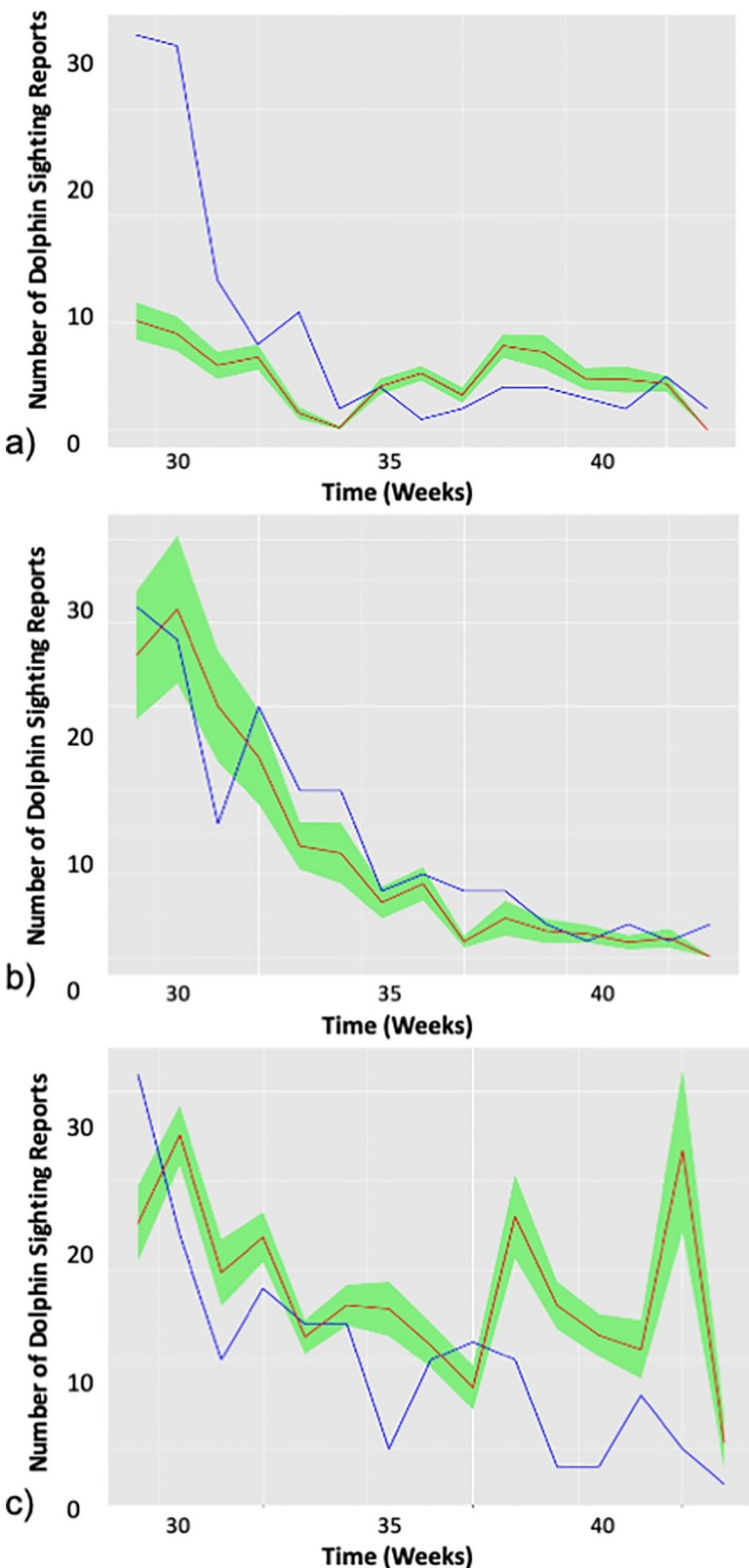

**Fig 7. Predicted versus actual bottlenose dolphin sightings.** Number of bottlenose dolphin sightings predicted using the final generalized additive models (red line) and the confirmed Chesapeake DolphinWatch sightings during between June and October 2019 (blue line) in the Upper (a), Middle (b), and Lower (c) Bay segments. Standard error

of predictions are shaded in green. The root mean square error (RMSE) and results of the Spearman's Rank Correlation test are listed for each Bay segment.

spot, and other fish species in the Chesapeake Bay, are euryhaline, meaning that they are able to adapt to a wide range of salinities [34], such as those in the Bay. The availability of euryhaline prey species may explain the distribution of dolphins throughout the Bay in a wide range of salinities (4 PPT—22 PPT).

Our study indicates that tidal states within the Chesapeake Bay may influence the observed spatiotemporal pattern of bottlenose dolphins in the Lower and Middle Bay. Coastal water levels are higher than average and water currents are faster during spring tidal states, creating optimal foraging regions for dolphins near the coastline [13]. During these spring tide conditions, smaller fish far from the coast may be swept inshore by the tide, increasing local prey abundance and diminishing the time and energy dolphins spent searching for food. Additionally, dolphins may take advantage of differential acoustic propagation during high tidal states [35–37]. These hypotheses require further exploration.

In the Upper Bay, however, tidal phase was not significantly related to the number of dolphin sightings. Tides in the Lower and Middle Bay ranged from approximately 0.9–1.3 meters during our study period, whereas in the Upper Bay segment, they only varied up to 0.8 meters (NOAA Tides and Currents data portal, tidesandcurrents.noaa.gov). The lower tidal variation may have had a smaller effect on prey availability and catchability, and thus no resulting increase in dolphin occurrence.

This study utilized sightings data gathered through citizen science. This method allowed us to noninvasively collect data on the locations of bottlenose dolphins across the entire Chesapeake Bay over three consecutive years. Though there were financial costs associated with app development and maintenance, we obtained species presence data at a relatively low cost. By quality controlling the dataset through validation of sighting reports, exclusion of duplicate dolphin sightings, and systematic confirmation with acoustic occurrence data, we are confident that these data provide a representative depiction of bottlenose dolphin presence at the scale of the three segments of the Bay.

While stranding records indicate that other marine mammals, such as harbor porpoise (*Phocoena phocoena*) and harbor seal (*Phoca vitulina*) do occur in the Chesapeake Bay, they are highly unlikely to occur during the summer study period [38, 39]. Therefore, the presence of ambiguous species identification for sightings was very low. This fact, in addition to the corresponding photos and/or videos and/or descriptions included in all sightings, provides reassurance that these sightings from Chesapeake DolphinWatch were indeed bottlenose dolphin detections.

Outside of the Chesapeake Bay, other projects have characterized marine mammal presence using citizen science interfaces. Whale mAPP, an application designed to study marine mammals around southeast Alaska, reported that their implementation of an opportunistic sightings database not only collected sufficient data, but also encouraged members of the community who were involved with the data collection process to self-initiate further learning in marine science [40]. Likewise, members of the Chesapeake DolphinWatch volunteer community have explored the marine environment in their own neighborhoods by walking along the coastline or participating in recreational activities such as boating, fishing, or paddleboarding while engaging in local scientific research.

Additionally, we communicated with citizen scientists through social media, email, and in-person events at local nature societies, sailing associations, and marinas. Residents of the Chesapeake Bay watershed have shown immense enthusiasm and appreciation for local wildlife as

well as efforts to protect and restore the estuary. Though a relatively high level of outreach has been attained with the Chesapeake DolphinWatch user network, the immense number of registered users does limit the degree of connection between them and the Chesapeake Dolphin-Watch team. The opportunistic-based observation approach used by this mobile application differs from other volunteer programs, which usually incorporate some degree of species identification training into their sighting reports. Quality control of the observations database as well as future deployments of passive acoustic monitoring devices throughout the Chesapeake Bay will be used to support the monitoring of local bottlenose dolphins.

This study offers insight into the seasonal presence of dolphins within the Chesapeake Bay, which is crucial to ecological management. The U.S. Marine Mammal Protection Act requires activities that may adversely affect marine mammals to receive permits. It also requires that the U. S. National Marine Fisheries Service (NMFS) ensure that all anthropogenic activities authorized by the Service have the least practicable adverse impact to local impacted marine mammals [41]. Previously, NMFS has not considered bottlenose dolphins as regular inhabitants of the Chesapeake Bay or required their inclusion within Environmental Impact Statements related to proposed developments and activities within the Bay that could cause harm or disturbance to the dolphins. When data is scarce, NMFS is required to take a precautionary approach, assuming that marine mammals would be present. This precautionary approach may result in increased costs or restrictions to activities, which ultimately do little to nothing to protect the animals. Our data and models on the spatiotemporal distribution of bottlenose dolphins should greatly assist in identifying times and locations that will minimize the impact to bottlenose dolphins and when mitigation measures would be most beneficial. For example, mitigations imposed on construction projects in the Chesapeake Bay in the month of October may be costly with little benefit provided to the population as most animals will have left the Bay by then whereas avoiding scheduling any potentially harmful activities in the peak months of June to August could have the most benefit.

Due to the Bay's high levels of marine traffic, planned construction, and military activity, managers who are tasked with monitoring the health of marine mammal populations can now use this information to minimize risk to this species. Because our models allow predictions of dolphin occurrence based on environmental conditions, these abiotic parameters can be used to infer when and where dolphins will occur in any time period. Our data and models could be used to study the potential overlap between dolphin occurrence and anthropogenic activities, such as vessel traffic and fishing. They could also be studied in relation to different habitat types that have been degraded or restored, such oyster reefs and submerged aquatic vegetation (underwater grasses).

This study, the first description of the spatiotemporal distribution of bottlenose dolphins within the Chesapeake Bay, provides a baseline from which future patterns of occurrence can be compared. Additional collection of acoustic and environmental data would provide context to the sightings made by citizen scientists and aid in determining the behaviors, including foraging, of dolphins in the Bay. This behavioral context would improve both the ecological understanding and management of bottlenose dolphins within the Chesapeake Bay. This study is the first description and model of the spatiotemporal distribution of bottlenose dolphins within the highly urbanized Chesapeake Bay. These findings can be used by resource managers to minimize the impacts of the many current and proposed anthropogenic activities in this region.

## Supporting information

**S1 Fig. Chesapeake DolphinWatch user Google Analytics data from both web and mobile-based application sources.**
(TIF)

**S1 File. Confirmation criteria for Chesapeake DolphinWatch dolphin sighting reports.**
(DOCX)

## Acknowledgments

Thank you to all of the staff and students who have assisted with this project at the Chesapeake Biological Laboratory. We particularly thank the citizen scientists who contributed their dolphin sightings to the Chesapeake DolphinWatch database, without whom this study would not have been possible.

## Author Contributions

**Conceptualization:** Lauren Kelly Rodriguez, Amber D. Fandel, Helen Bailey.

**Data curation:** Lauren Kelly Rodriguez, Jamie C. Testa.

**Formal analysis:** Lauren Kelly Rodriguez.

**Investigation:** Lauren Kelly Rodriguez.

**Methodology:** Amber D. Fandel, Helen Bailey.

**Project administration:** Helen Bailey.

**Resources:** Lauren Kelly Rodriguez, Amber D. Fandel, Benjamin R. Colbert, Jamie C. Testa, Helen Bailey.

**Supervision:** Amber D. Fandel, Benjamin R. Colbert, Helen Bailey.

**Validation:** Jamie C. Testa.

**Writing – original draft:** Lauren Kelly Rodriguez, Helen Bailey.

**Writing – review & editing:** Lauren Kelly Rodriguez, Amber D. Fandel, Benjamin R. Colbert, Jamie C. Testa, Helen Bailey.

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
