## [Decision Letter · Decision Letter 0]

8 Oct 2020

PONE-D-20-27115

Spatial and temporal variation in the occurrence of bottlenose dolphins in the Chesapeake Bay using citizen science sighting data

PLOS ONE

Dear Dr. Rodriguez,

Thank you for submitting your manuscript to PLOS ONE. After careful consideration, we feel that it has merit but does not fully meet PLOS ONE’s publication criteria as it currently stands. Therefore, we invite you to submit a revised version of the manuscript that addresses the points raised during the review process.

An expert in the field has reviewed the manuscript and found the work to be of interest. Detailed comments to improve the paper are included in the attached document. Please correct the grammatical mistakes and provide more explanation for the variables examined. Particular attention should be paid to potential bias in observer measurements.

We look forward to receiving your revised manuscript.

Kind regards,

Cheryl S. Rosenfeld, DVM, PhD

Academic Editor

PLOS ONE

Journal Requirements:

3. Please upload a new copy of Figures 1, 4, and 5 as the detail is not clear. Please follow the link for more information: https://blogs.plos.org/plos/2019/06/looking-good-tips-for-creating-your-plos-figures-graphics/" https://blogs.plos.org/plos/2019/06/looking-good-tips-for-creating-your-plos-figures-graphics/

4.  We note that [Figures  1, 4 and 5] in your submission contain [map/satellite] images which may be copyrighted. All PLOS content is published under the Creative Commons Attribution License (CC BY 4.0), which means that the manuscript, images, and Supporting Information files will be freely available online, and any third party is permitted to access, download, copy, distribute, and use these materials in any way, even commercially, with proper attribution. For these reasons, we cannot publish previously copyrighted maps or satellite images created using proprietary data, such as Google software (Google Maps, Street View, and Earth). For more information, see our copyright guidelines: http://journals.plos.org/plosone/s/licenses-and-copyright.

1.    You may seek permission from the original copyright holder of Figure(s) [1, 4 and 5] to publish the content specifically under the CC BY 4.0 license.  

Reviewers' comments:

Reviewer's Responses to Questions

**Comments to the Author**

1. Is the manuscript technically sound, and do the data support the conclusions?

Reviewer #1: Yes

2. Has the statistical analysis been performed appropriately and rigorously? 

Reviewer #1: Yes

3. Have the authors made all data underlying the findings in their manuscript fully available?

Reviewer #1: Yes

4. Is the manuscript presented in an intelligible fashion and written in standard English?

Reviewer #1: Yes

5. Review Comments to the Author

Reviewer #1: Congratulations to the authors on the preparation of the manuscript. I recommend that the paper is accepted for publication with major review. Please see specific comments in the attached document.

Kind regards,

Reviewer.

6. PLOS authors have the option to publish the peer review history of their article (what does this mean?). If published, this will include your full peer review and any attached files.

Reviewer #1: No

---

## [Author Response · Author response to Decision Letter 0]

13 Dec 2020

November 22, 2020

General Comments:

Authors present data on dolphin sightings in Chesapeake Bay, USA using citizen science (CS) data collected over three years. The research provides an important baseline dataset to inform conservation management and impact assessment into the future. It is recommended that the authors provide additional information on the observer bias and effort from CS sightings and how this affected the results. 

We have made sure to add detail regarding potential observer bias in lines 364-365, 415-442, and 2155-2163. Further details regarding how bias and effort from CS sightings was accounted for in our analysis is given in our responses to specific comments below. 

Lines 364-365: “Summing the observational reports per week reduced temporal sampling bias caused by higher reporting rates on weekend days.”

Lines 415-442: “The Chesapeake DolphinWatch citizen science project provided a network of observers over a large spatial extent of the Bay, but its unsystematic nature may have 

affected the temporal patterns observed in the sightings. People are generally more active 

on the water during summer months in the study area, which could create temporal bias in 

the citizen science reports. We therefore used acoustic detections of bottlenose dolphins to verify the seasonal occurrence pattern observed.”

Lines 2155-2163: “Though a relatively high level of outreach has been attained with the Chesapeake DolphinWatch user network, the immense number of registered users does limit the degree of connection between them and the Chesapeake DolphinWatch team. The opportunistic-based observation approach used by this mobile application differs from other volunteer programs, which usually incorporate some degree of species identification training 

into their sighting reports. Quality control of the observations database as well as future 

deployments of passive acoustic monitoring devices throughout the Chesapeake Bay 

will be used to support the monitoring of local bottlenose dolphins.”

The paper is well written. Some sections could use some restructuring and clarity. The analysis supports the objectives and results. Some justification could be further provided on selection of variables i.e. why was depth not considered? 

The models implemented in this study only accounted for three broad spatial regions. In examining this, we decided that averaging the shallow depths (~4-7 meters) of the tributaries with the deeper depths of the mainstream (~25-50 meters) would not be informative to our study. We have added how we addressed our variable selection process, specific to depth, in our response to specific comments below. 

The author is encouraged to define the seasons and survey period more clearly throughout the manuscript. 

We have added a defined survey period to our manuscript in the same section where we define the study site (lines 350-353). Data for this study was collected from June 2017 through October 2019. However, only the months April through October of each year (2017-2019) were analyzed due to the seasonal pattern of bottlenose dolphin occurrence throughout the study site and to reduce the effect of lower participation by citizen scientists during the winter. 

Lines 350-353: “Temporally, data (regarding species presence and environmental conditions) were collected from June 28, 2017 to October 14, 2019. Only data from April through October of each year were examined for this analysis. This does exclude the months of April, May, and June of 2017, in which data collection did not begin until late June.”

Specific Comments:

Title: 

Can you insert Chesapeake Bay, USA?

We have added “USA” to the title to clarify the location of this study. 

Abstract:

To avoid repetition you could remove ‘to determine relative occurrence throughout the estuary’

We have removed this sentence fragment from line 9 as suggested. 

Were CS sightings recorded all year round? Please clarify

The Chesapeake DolphinWatch app was offline for upgrades during December 2017 - March 2018. CS sightings could be recorded all year from April 2018, however, the data used in our analysis were from April-October only (line 8) because use of the app dramatically declined during the winter months and bottlenose dolphins are seasonal migrants who have only occasionally been documented in the Chesapeake Bay during winter months (referenced in paper: Engelhaupt et al., 2016). 

Was the acoustics data collected opportunistically, through CS as well?

Acoustic data were collected systematically through moored, archival dolphin echolocation click detectors, not through CS. We have clarified this in lines 11-12. 

Lines 11-12: “This pattern of observed occurrence was confirmed with systematic, passive acoustic detections of dolphin echolocation clicks from hydrophones

Introduction:

Line 71: Insert United States (U.S.), same for first use of USA.

We have added “United States” to the first description of our study location (line 94). 

Line 97: Results/ sample size/No. sightings should be reported in the results section, not introduction 

We have moved the sample size and number of sightings to the results section (lines 883-887) as suggested.

Methods:

Line 145: remove ‘their’

“Their” was removed from line 339 as suggested. 

Line 153: was there effort bias in different months i.e. sunny days in summer vs cold winter days. How did the number of observers vary? Was effort recorded when no sightings were recorded?

We clarify effort bias in lines 893-946, in which winter days within the study period (2017-2019) received little to no bottlenose dolphin sighting reports. We observed a lower number of application users (observed in Google Analytics user reports; Supplemental Figure 1) as well as less frequent dolphin sighting reports during winter months. We assessed the variation of CS effort by including Chesapeake DolphinWatch daily user frequency in each GAM (Lower, Middle, Upper) as an explanatory variable (lines 734-736). This user data is shown as a frequency distribution in Figure S1. 

Lines 893-946: “The week with the fewest sighting reports was also the same in each year – the final week of October (week 44; Fig. 3a). We focused our study analysis on data from April to October of each year as the application was offline during winter 2017-2018 and sighting reports were very low during winter 2018-2019, which could have resulted from less outdoor activity by our citizen science network.“

Lines 734-736: “The total number of Chesapeake DolphinWatch application users per week was included as an explanatory variable to offset the number of sightings and account for seasonal variation in user frequency.”

There was no specific effort reporting in our citizen science database, which reflected species presence data rather than presence/absence data. The nature of the mobile app meant that we were engaging a large audience of thousands of observers, but consequently there was no in-person training and hence quantifying effective effort is extremely difficult. We utilized the Google analytics mobile app daily user reports as a proxy for effort to help explain some of the temporal variation in dolphin observations. 

Line 155: assume this is the location of the observer, not the dolphin. Did you calculate an error range, what was the distance of dolphins from the observer? Was there other species in the Bay? Were photographs used to confirm sighting records/species? Was there any training completed for CS to have a level of competency when recording sightings data, species, group size etc?

There was no calculated error range for each reported sighting location. We accounted for variation in location reports by generalizing the spatial extent to three broad segments of the Chesapeake Bay. Due to the thousands of reports we received via Chesapeake DolphinWatch, it is challenging to account for location error that would vary based on the observer, their platform, and the accuracy of their mobile GPS. If the reporter(s) were following safe dolphin-watching guidelines (https://www.fisheries.noaa.gov/national/marine-life-viewing-guidelines/protect-wild-dolphins-admire-them-distance), then they were 50 meters or further from any dolphin. 

We clarify other marine mammal presence in the Bay on lines 1544-1613. When available, photos and video were used to confirm sighting reports of bottlenose dolphins. 

Lines 1544-1613: “While stranding records indicate that other marine mammals, such as harbor porpoise (Phocoena phocoena) and harbor seal (Phoca vitulina) do occur in the Chesapeake Bay, they are highly unlikely to occur during the summer study period (Aschettino, Engelhaupt et al. 2016, Hayes, Josephson et al. 2018). Therefore, the presence of ambiguous species identification for sightings was very low. This fact, in addition to the corresponding photos and/or videos included in all sightings, provides reassurance that these sightings from Chesapeake DolphinWatch were indeed bottlenose dolphin detections.”

Citizen scientists were not trained for species identification. The goal of the Chesapeake DolphinWatch application is to engage a large network of dolphin reporters across the Chesapeake Bay. With over 7,000 registered users, training would require more time and money than we are able to spend. Because dolphins are an easily recognizable species and we conducted quality control on our database of sightings, we were reassured that our reported sightings were of bottlenose dolphins. 

Group size was not considered in this analysis due to lack of confidence that untrained citizen scientists were reliably able to accurately identify the number of individuals within a dolphin pod. 

Line 177: Did you consider duplicates of the same dolphin group in a day, i.e. sightings of the same group in different areas >30m apart 

Unfortunately, it was impossible to differentiate dolphin groups within the same day based on current data. The sighting reports from the citizen scientists did not provide sufficiently detailed information to know if they were dolphins from the same group and it has been observed that dolphins form large groups and then sub-groups regularly within the Bay.

Line 192: perhaps add, during the 6-month period that the loggers were deployed each year. How did you consider observer bias outside of the 6-month period the loggers were deployed?

C-PODs were deployed for six months (May-October), which only excludes one month of our analyzed study period (April-October). We carefully selected the study months based on two observations: 

More people are active in and around the Chesapeake Bay during these months. This was confirmed by the increased mobile application views during April to October (see Supplemental Figure 1).

Previous literature indicates that bottlenose dolphins are most likely to occur in the study area during these months (referenced in paper: Barco et al. 1999, Toth et al. 2011, Gannon and Waples, 2004, Hayes et al. 2018). 

Results:

Line 277: what was the criteria for ‘confirming’ or excluding the sightings data?

All sightings entered into the app are automatically set to ‘unconfirmed’ status by the software. When a sighting has been reviewed by a staff member it is confirmed under the following criteria:

there is a photo or video of the encounter to accompany the sighting details provided and the sighting location is in the water

Or 

there is a detailed description of the sighting and the sighting was located in a portion of the Bay or its tributaries where previous sightings have been documented 

 Or

the user is part of a small list of “trusted application users” (n=14) who have previously demonstrated competence in identifying bottlenose dolphins in the Chesapeake Bay multiple times through detailed descriptions and images (photos/videos) 

If a sighting location is on land and contains a photo/video and/or written description, staff will make an attempt to obtain the correct location information by emailing the user. If there is no response to our email, the sighting remains unconfirmed for the season and is ‘rejected’ at the end of the year. If the location, time, group size, etc. of the sighting does not match the description written by the user, staff would email the user to clarify the data. If the sighting occurs in an area not previously known to have dolphin sightings and does not have a photo/video to corroborate, it remains unconfirmed, unless the user responds to our email and sends additional information (i.e. a detailed description or photo/video) to confirm the sighting.

We address criteria for confirming the sighting reports in lines 378-387. We have also added an additional supplement (Supplement 1) which describes the process in further detail. 

Lines 378-387: “Application users were required to register and login with their email address so that they could be contacted for further information about their sighting. For a reported sighting to be confirmed as a true sighting, a detailed description and/or an original, verified photo or video of the event was necessary. If a description, photo, or video was not included in the report, the user was contacted via email with a request for additional details about their sighting. Some sightings from “trusted application users,” a small group of people who had been in contact with our team and regularly submitted reliable sighting reports, were confirmed even if no description or photographic evidence was included. Further details on the criteria for confirming or excluding sighting reports is included in Supplement 1.”

Line 279: Please define whether you are talking about inter or intra seasonal variation. I suggest restructuring text and separating the paragraphs here to report results for a) inter-seasonal variation b) intra-seasonal variation within years including peak numbers and c) spatial variation inter and intra seasonal.

To clarify, we only examined the period April-October of each year in our analysis, we were not examining the entire year. Therefore, we believe that users may be confused with the use of “inter-seasonal variation” because the summer season makes up the bulk of our study period. We have clarified “interannual” and “intraseasonal” results on lines 888 and 890. 

Lines 888-890: “There was interannual variation in bottlenose dolphin sightings (Figure 3a). More sightings occurred in all three segments of the Bay in 2018 (Table 1), though each year showed a similar intraseasonal pattern of sightings.”

Line 310: were the CS land based or boat based? Assuming that the location of the observer was recorded, not the dolphins, what is your confidence for the location of sightings?

CS sightings were both land and boat-based. Bottlenose dolphins can swim relatively close to the shore. Due to this and their relatively large body size, citizen scientists are easily able to identify their pods from land. 

The accuracy of the location of sightings will vary with the observer, how quickly they reported their sighting after the observation, and the GPS accuracy and cellular reception of their mobile platform at the time of the report. It is therefore extremely difficult to quantify our confidence in sighting locations for over a thousand observers, therefore we generalized our study areas to three latitudinally-equal segments of the Bay. 

Line 314: was highest sightings in this area because of concentrated effort?

We have taken a closer look at this area, the mouth of the Rappahannock River. Many sightings in this area were reported by one individual, which we have explained on lines 1005-1010. We plan to deploy a CPOD in this area during future summer seasons to further explore the spatially-explicit occurrence patterns of bottlenose dolphins. 

Lines 1005-1010: “The highest frequency (n = 136) of sightings was located at the mouth of the Rappahannock River (37.57 °N, -76.34 °W) in the Middle Bay. The majority (74%) of these reports were submitted to the app by a single observer. Note that we were unable to test if the Rappahannock River was truly the most significant location for dolphin sightings or if this user’s diligent efforts created spatial bias.”

Can you assess statistic difference between no of sightings in different areas and if the occurrence was statistically different in different zones, insert p-value?

Thank you for this suggestion. We have applied an ANOVA to assess the difference between the weekly number of dolphin sightings and the area of the Bay (Lower, Middle, Upper). The sighting frequency was not statistically different between the different segments (F2,231 = 1.629, p = 0.198). We have added this statistic to the manuscript in line 760. 

Discussion:

Line 384: author is encouraged to define the seasons and survey period more clearly throughout the manuscript

We apologize for any confusion regarding our study period. We have added the defined study period to lines 328-331 to help clarify this information to readers. 

Line 328-331: “Data for this study was collected from June 28, 2017 to October 14, 2019. Only data from the months April to October of each year were examined for this analysis.”

The first paragraph of the discussion repeats the results. Can authors please re-phrase as an interpretation of results to tie into discussion points. Could try focusing on topic sentences and then backing up with a summary of result to back up point.

We appreciate this recommendation to improve the first paragraph of the discussion and have rewritten it to reflect the reviewer’s suggestions. 

“Bottlenose dolphins have not previously been considered to be regular visitors of the Chesapeake Bay. Our study described the spatial and temporal occurrence of dolphins, which was relatively consistent across our three years of study. The heightened seasonal presence of dolphins in this area was related to water temperature and salinity, two environmental parameters that are continuously and systematically measured in the Bay. As an urbanized, coastal region, it’s crucial for species management in and around the Chesapeake Bay to be able to assess the spatiotemporal occurrence of protected species, such as bottlenose dolphins. We observed a distinct pattern of dolphin sightings during the summer season, that should be taken into account to take appropriate measures to mitigate the effects of anthropogenic activity which may harm or disturb bottlenose dolphins.”

Line 426: I wonder if other factors influence dolphin presence in an area in high tide, such as more water and therefore room to swim, rest, socialise (what was the water depth in these area), also there may be benefits acoustically during higher tide, less rapid attenuation of calls and therefore more successful fishing? Just thoughts – perhaps the author could explore some other potential hypothesis.

We appreciate these suggestions regarding the factors influencing dolphin preference for areas with high tide. Previous literature does indicate that tidal phase can affect acoustic propagation (referenced in paper: Best et al. 2004, Hamilton and Bachman 1998, Willis et al. 2013), therefore, we have included this and other potential hypotheses in the discussion section on lines 1279-1529.

Lines 1279-1529: “Our study indicates that the tidal state within the Chesapeake Bay may influence the observed spatiotemporal pattern of bottlenose dolphins in the Lower and Middle Bay. Coastal water levels are higher than average and water currents are faster during spring tidal states, creating optimal foraging regions for dolphins near the coastline (Fury and Harrison 2011). During these spring tide conditions, smaller fish far from the coast may be swept inshore by the tide, increasing local prey abundance and diminishing the time and energy dolphins spent searching for food. Additionally, dolphins may take advantage of differential acoustic propagation during high tidal states (Hamilton and Bachman 1982, Best, Tuffin et al. 2004, Willis, Broudic et al. 2013). These hypotheses require further exploration.”

Water depth would be a good variable to include in analysis

The models implemented in this study only accounted for three broad spatial regions. In examining this, we decided that averaging the shallow depths (~4-7 meters) of the tributaries with the deeper depths of the mainstream (~25-50 meters) would not be informative to our study. 

Line 437: why are other species highly likely to occur in the summer period? Can you provide references?

There may be a misunderstanding with line 1546 - It’s written that other species are highly unlikely to occur in the summer period. We have highlighted this part in the text and provided additional references. 

Author should identify the limitations with data and room for improvement i.e. bias with observers and effort and level of training.

We appreciate this suggestion and have added lines 1632-1634 to explain our planned future improvements to the Chesapeake DolphinWatch database. 

Lines 1632-1634: “Quality control of the observations database as well as future deployments of passive acoustic monitoring devices throughout the Chesapeake Bay will be used to support the monitoring of local bottlenose dolphins.”

We recognize that the remote level of interaction amongst our large group of over 7,000 users means we are unable to provide training across all of the citizen scientists. Opportunistic, species-presence data will always be affected by spatiotemporal bias due to inherent variation in observer activity. We plan in the future on deploying acoustic recorders in additional regions of the Chesapeake Bay as well as throughout the winter season to account for the reviewer’s suggestion to analyze interseasonal bias in bottlenose dolphin occurrence. 

Due to the potential for some errors in the location accuracy of sighting reports (even after quality control procedures), we were conservative and did not seek in this study to analyze dolphin sighting patterns at a finer spatial scale.

December 13, 2020

1) Thank you for stating in your methods that: "A C-POD (Chelonia Wildlife Acoustic Monitoring, www.chelonia.co.uk) was deployed in the middle Bay at the mouth of the Potomac River (37.914 °N, 76.258 °W) from May through October during each study year (Fig 1a)."

To comply with PLOS ONE submissions requirements for field studies, please provide the following information in the Methods section of the manuscript and in the “Ethics Statement” field of the submission form (via “Edit Submission”):

a) Provide the name of the authority who issued the permission for each location (for example, the authority responsible for a national park or other protected area of land or sea, the relevant regulatory body concerned with protection of wildlife, etc.). If the study was carried out on private land, please confirm that the owner of the land gave permission to conduct the study on this site.

The C-POD used in this study was not deployed in a park, sanctuary, within a navigational channel, or in any other protected area which would require regulatory approval. Additionally, the study relied solely on passive data collection and therefore does not require a permit under the U.S. Endangered Species Act, the Marine Mammal Protection Act, or any state's equivalent natural resource law.

b) For any locations/activities for which specific permission was not required, please

- i. state clearly that no specific permissions were required for these locations/activities, and provide details on why this is the case

- ii. confirm that the field studies did not involve endangered or protected species

This study involved the passive acoustic monitoring of the protected species, bottlenose dolphins and thus, no specific permissions were required.

c) For vertebrate studies only, please provide the following additional information:

- i. Full details of collection and sampling methods, including method of sacrifice if applicable

- ii. State whether the vertebrate work was approved by an Institutional Animal Care and Use Committee (IACUC) or equivalent animal ethics committee. If no approval was obtained, please explain why it was not required.

- iii. State clearly whether all sampling procedures and/or experimental manipulations were reviewed or specifically approved as part of obtaining the field permit.

While the goal of this study was to study vertebrate ecology, the data collection solely utilized a passive acoustic recorder. There was physical interaction with, collection of, or sacrifice of marine mammals or other vertebrates for this research. Therefore, no IACUC approval was required, nor were permits or authorizations under any natural resources law.

2) We note that you have resubmitted new copies of Figs 1, 4, and 5, but you have not addressed the copyright concerns for these images. Please respond to our concerns, listed again below:

Figures 1, 4, and 5 utilize the World Light Gray Reference basemap available with ArcGIS 10.6. This basemap was compiled by Esri using HERE data, Garmin basemap layers, OpenStreetMap data, GIS community data, and Esri basemap data. As indicated by the Esri Master License Agreement (https://www.esri.com/content/dam/esrisites/en-us/media/legal/ma-translations/english.pdf, https://downloads2.esri.com/arcgisonline/docs/tou_summary.pdf), this basemap is allowed to be printed in academic publications and does not need further permission from the copyright holder. We have included additional attribution to Esri and its data providers in the captions for Figs 1, 4, and 5. 

3) We note that your manuscript is not formatted using one of PLOS ONE’s accepted file types. Please reattach your manuscript as one of the following file types: .doc, .docx, .rtf, or .tex (accompanied by a .pdf).

If your submission was prepared in LaTex, please submit your manuscript file in PDF format and attach your .tex file as “other.”

Our manuscript file has been reuploaded as a .docx file labeled “Manuscript.docx”. 

4) Please amend the title either on the online submission form or in your manuscript so that they are identical.

We have ensured that each part of the online submission form as well as all uploaded copies of the manuscript share identical titles.

---

## [Decision Letter · Decision Letter 1]

31 Mar 2021

PONE-D-20-27115R1

Spatial and temporal variation in the occurrence of bottlenose dolphins in the Chesapeake Bay, USA, using citizen science sighting data

PLOS ONE

Dear Dr. Dr. Rodríguez,

Thank you for submitting your manuscript to PLOS ONE. After careful consideration, we feel that it has merit but does not fully meet PLOS ONE’s publication criteria as it currently stands. Therefore, we invite you to submit a revised version of the manuscript that addresses the points raised during the review process.

We look forward to receiving your revised manuscript.

Kind regards,

Susana Caballero, PhD

Academic Editor

PLOS ONE

Journal Requirements:

Additional Editor Comments (if provided):

Thank you for carefully working on all suggestions made by the reviewers on the previous version of the manuscript. The reviewer congratulates you on improving the quality and clarity of your manuscript. There are some minor points to work on now, particularly improving the clarity of some of the criteria used in accepting which sightings to include in your analyses. Also expanding a bit in the introduction.

Reviewers' comments:

Reviewer's Responses to Questions

**Comments to the Author**

1. If the authors have adequately addressed your comments raised in a previous round of review and you feel that this manuscript is now acceptable for publication, you may indicate that here to bypass the “Comments to the Author” section, enter your conflict of interest statement in the “Confidential to Editor” section, and submit your "Accept" recommendation.

Reviewer #1: All comments have been addressed

2. Is the manuscript technically sound, and do the data support the conclusions?

Reviewer #1: Yes

3. Has the statistical analysis been performed appropriately and rigorously? 

Reviewer #1: Yes

4. Have the authors made all data underlying the findings in their manuscript fully available?

Reviewer #1: Yes

5. Is the manuscript presented in an intelligible fashion and written in standard English?

Reviewer #1: Yes

6. Review Comments to the Author

Reviewer #1: The manuscript has been improved and authors are congratulated on addressing reviewer feedback. Please see minor revisions and comments below. It is recommended that the manuscript is accepted with minor revisions.

Abstract:

Can the last sentence be broadened? Can the models can be used for predictive tool for species occurrence in important cetacean habitat, including Chesapeake Bay?

Methods/Results:

The bias and potential error in location data is explained in the response to reviewers, but it is not clear where it is described in the manuscript text. Please update the manuscript to include a description of the biases raised in the first review. Is there a protocol that requires the citizen scientists to report the sighting at the location of sighting? What measures are in place to prevent them from carrying on with their day and then reporting the sighting at a different time and location?

More clarity should be provided in the manuscript on the acceptance criteria for ‘accepting’ sightings. I understand that supplementary material is provided, but I think the manuscript would benefit from a short description.

Table 1: total sightings reports for 2017-2019 add up to 2,907. This is inconsistent with the text that states that a total of 2,759 sightings were reported. Similarly Table 1 reports a total of 1,788 confirmed sightings and text states 1,870. Please cross check all values in text and figures throughout manuscript.

Discussion:

What are the current threats to dolphins in the Chesapeake Bay? As a suggested future study it would be important for conservation management to understand the spatial overlap between dolphin occurrence and human activities i.e. boating, fishing, water quality (is there contaminated water?), is habitat degradation a risk?

Is there any current management protection for the cetaceans in the Bay? Can you provide some recommendations to management?

Line 414-422: please revise this paragraph. The explanation of the precautionary principle doesn’t make sense to me.

Line 416: should this be prohibited, not ‘permitted’? Please revise sentence.

Line 417: is practical effect the correct terminology here? I would think that ‘potential effect’ is more suitable.

Line 427: I think ‘determine’ should be replaced with ‘infer’ or ‘indicate’ when are where dolphins ‘may occur seasonally’.

Last sentence: I think the last sentence could be more punchy.

7. PLOS authors have the option to publish the peer review history of their article (what does this mean?). If published, this will include your full peer review and any attached files.

Reviewer #1: No

---

## [Author Response · Author response to Decision Letter 1]

23 Apr 2021

1) The manuscript’s reference list was reviewed to ensure all appropriate citations were included. One reference (#23: R Core Team. R: A language and environment for statistical computing. R Foundation for Statistical Computing. 2017.) was added to the reference list in this version of the manuscript after being left out in prior versions.

In-text line 195-198: To determine whether the acoustic detections and reported sightings showed similar trends in dolphin presence, a Spearman’s Rank Correlation test in R using the cor.test function was utilized [23].

Abstract:

Can the last sentence be broadened? Can the models can be used for predictive tool for species occurrence in important cetacean habitat, including Chesapeake Bay?

The scope of this paper is to use our Generalized Additive Models as a predictive tool for this species’ (bottlenose dolphins, Tursiops truncatus) occurrence in the Chesapeake Bay. As it is the first paper of its kind in this urban region, it’s noteworthy information to share these models with management and other regulatory agencies in the area. However, the response of bottlenose dolphins to the environmental conditions of this Mid-Atlantic system may differ from the response of dolphins in other regions that have very different temperature and salinity regimes, such as that of the Gulf of Mexico or off California. Therefore, we only recommend using these models for predictive capabilities in the Chesapeake Bay, although similar statistical modeling approaches could be used in other locations. 

Methods/Results:

The bias and potential error in location data is explained in the response to reviewers, but it is not clear where it is described in the manuscript text. Please update the manuscript to include a description of the biases raised in the first review. Is there a protocol that requires the citizen scientists to report the sighting at the location of sighting? What measures are in place to prevent them from carrying on with their day and then reporting the sighting at a different time and location?

The sighting location can be based on the device’s GPS location if reported at the time of the sighting or by selecting a point on the map if the user enters the sighting afterwards or they saw the dolphins from the shore. If the sighting location is plotted on land or if it does not match the description entered, the sighting was not confirmed and not included in our final database for the analyses. We have added this information on lines 89-98, 120-126, and 148-152. 

In-text lines 89-98: To account for variability in environmental conditions throughout the estuary, and because the potential location error in the sightings reported by our citizen scientists was unknown, we spatially divided the Chesapeake Bay in this study into three latitudinally-equal segments (Upper, Middle, Lower) with the Lower segment being the one closest to the Atlantic Ocean (Fig 1). 

In-text lines 120-126: Each sighting report entered into Chesapeake DolphinWatch required the location of the sighting, which could be obtained directly from the user’s device or by the user selecting a point on a map. The time and date of the sighting were also recorded. A section for additional details on the sighting allowed users to include a description and/or pictures and videos of what they observed (Fig 2). Users could enter this information during the sighting or afterwards.

In-text lines 148-152: Users periodically enter sightings with insufficient information to be considered a true sighting and in these instances the data remained in the application but are deemed “unconfirmed”. Sighting locations that did not match the entered description or that occurred on land were not confirmed. Unconfirmed sightings were not included in this analysis. 

More clarity should be provided in the manuscript on the acceptance criteria for ‘accepting’ sightings. I understand that supplementary material is provided, but I think the manuscript would benefit from a short description. 

We have included additional detail on the acceptance criteria that we used to confirm citizen science dolphin sightings in the manuscript from lines 142-152. 

In-text lines 142-152: For a reported sighting to be confirmed as a true sighting, it must meet the following requirements: 1) there is a photo or video of the encounter to accompany the sighting details provided and the location of the sighting is plotted in the water, or 2) there is a description of the sighting event. Some sightings from “trusted application users,” a small group of people who were in contact with our team and regularly submitted reliable sighting reports, were confirmed even if no description or photographic evidence was included. Users periodically enter sightings with insufficient information to be considered a true sighting, and in these instances the data remain in the application but are deemed “unconfirmed”. Sighting locations that did not match the entered description or that occurred on land were not confirmed. Unconfirmed sightings were not included in this analysis. 

Table 1: total sightings reports for 2017-2019 add up to 2,907. This is inconsistent with the text that states that a total of 2,759 sightings were reported. Similarly Table 1 reports a total of 1,788 confirmed sightings and text states 1,870. Please cross check all values in text and figures throughout the manuscript.

Thank you for catching this inconsistency. We have re-checked and updated all summary statistics throughout the manuscript.

Discussion:

What are the current threats to dolphins in the Chesapeake Bay? As a suggested future study it would be important for conservation management to understand the spatial overlap between dolphin occurrence and human activities i.e. boating, fishing, water quality (is there contaminated water?), is habitat degradation a risk?

The introduction of this manuscript (lines 35-41) describes some of the threats to dolphins in the Chesapeake Bay including those from 1) naval operations, 2) recreational boating, and 3) commercial fishing/shipping operations. We have added sentences on lines 384-385 and 478-481 in the discussion to emphasize the importance of these anthropogenic threats and habitat types in and around the Chesapeake Bay. 

In-text lines 35-41: The Chesapeake Bay is a large, highly urbanized estuary along the Mid-Atlantic coast of the USA. The Bay is characterized by tourism, military activities, and shipping; all of which can disturb marine species [3]. The Chesapeake Bay is home to two of the largest shipping ports in the USA (in Baltimore, Maryland and Hampton Roads, Virginia) as well as the largest naval base in the world (in Norfolk, Virginia). Dolphins that frequent this area are likely to encounter noise and physical disturbance from recreational and commercial shipping as well as from naval training exercises.

In-text lines 384-385: Between regional naval operations, urban coastal construction, and recreational activity, the Chesapeake Bay is a relatively loud and crowded marine environment.

In-text lines 478-481: Because our models allow predictions of dolphin occurrence based on environmental conditions, these abiotic parameters can be used to determine when and where dolphins will occur in any time period. Our data and models could be used to study the potential overlap between dolphin occurrence and anthropogenic activities, such as vessel traffic and fishing. They could also be studied in relation to different habitat types that have been degraded or restored, such oyster reefs and submerged aquatic vegetation (underwater grasses).

Is there any current management protection for the cetaceans in the Bay? Can you provide some recommendations to management?

The current management protection for bottlenose dolphins along the Mid-Atlantic region of the United States is the Marine Mammal Protection Act, which is not specific to the Chesapeake Bay. Prior to the launch of the Chesapeake DolphinWatch platform and publication of this paper, natural resource managers were not aware of, and therefore did not consider, the frequent and regular occurrence of bottlenose dolphins in the Chesapeake Bay. The recommendation that we have within this manuscript is that bottlenose dolphin presence, as a protected species, be incorporated in future Environmental Impact Statements, which are required for any construction or activity that may cause a major disturbance. This description has been expanded in the discussion on lines 457-475. 

In-text lines 457-475: The U.S. Marine Mammal Protection Act requires activities that may adversely affect marine mammals to receive permits. It also requires that the U.S. National Marine Fisheries Service (NMFS) ensure that all anthropogenic activities authorized by the Service have the least practicable adverse impact to local impacted marine mammals [40]. Previously, NMFS has not considered bottlenose dolphins as regular inhabitants of the Chesapeake Bay or required their inclusion within Environmental Impact Statements related to proposed developments and activities within the Bay that could cause harm or disturbance to the dolphins. When data is scarce, NMFS is required to take a precautionary approach, assuming that marine mammals would be present. This precautionary approach may result in increased costs or restrictions to activities, which ultimately do little to nothing to protect the animals. Our data and models on the spatiotemporal distribution of bottlenose dolphins should greatly assist in identifying times and locations that will minimize the impact to bottlenose dolphins and when mitigation measures would be most beneficial. For example, mitigations imposed on construction projects in the Chesapeake Bay in the month of October may be costly with little benefit provided to the population as most animals will have left the Bay by then whereas avoiding scheduling any potentially harmful activities in the peak months of June to August could have the most benefit.

Line 414-422: please revise this paragraph. The explanation of the precautionary principle doesn’t make sense to me.

This paragraph has been revised for clarity. 

In-text lines 456-475: This study offers insight into the seasonal presence of dolphins within the Chesapeake Bay, which is crucial to the ecological management of this region. The U.S. Marine Mammal Protection Act requires activities that may adversely affect marine mammals to receive permits. It also requires that the U.S. National Marine Fisheries Service (NMFS) ensure that all anthropogenic activities authorized by the Service have the least practicable adverse impact to local impacted marine mammals [40]. Previously, NMFS has not considered bottlenose dolphins as regular inhabitants of the Chesapeake Bay or required their inclusion within Environmental Impact Statements related to proposed developments and activities within the Bay that could cause harm or disturbance to the dolphins. When data is scarce, NMFS is required to take a precautionary approach, assuming that marine mammals would be present. This precautionary approach may result in increased costs or restrictions to activities, which ultimately do little to nothing to protect the animals. Our data and models on the spatiotemporal distribution of bottlenose dolphins should greatly assist in identifying times and locations that will minimize the impact to bottlenose dolphins and when mitigation measures would be most beneficial. For example, mitigations imposed on construction projects in the Chesapeake Bay in the month of October may be costly with little benefit provided to the population as most animals will have left the Bay by then whereas avoiding scheduling any potentially harmful activities in the peak months of June to August could have the most benefit.

Line 416: should this be prohibited, not ‘permitted’? Please revise sentence.

We have changed the wording of this sentence. 

In-text lines 457-459: The U.S. Marine Mammal Protection Act requires activities that may adversely affect marine mammals to receive permits.

Line 417: is practical effect the correct terminology here? I would think that ‘potential effect’ is more suitable.

The correct terminology in this case is "least practicable adverse impact". We have changed the wording of this sentence to make it clearer. 

In-text lines 459-461:. It also requires that the U.S. National Marine Fisheries Service (NMFS) ensure that all anthropogenic activities authorized by the Service have the least practicable adverse impact to local impacted marine mammals [40].

Line 427: I think ‘determine’ should be replaced with ‘infer’ or ‘indicate’ when are where dolphins ‘may occur seasonally’.

We agree that “infer” is a clearer describer than “determine”. We have made this change in the manuscript.

In-text lines 479-481: Because our models allow predictions of dolphin occurrence based on environmental conditions, these abiotic parameters can be used to infer when and where dolphins will occur in any time period.

Last sentence: I think the last sentence could be more punchy.

Thank you for this suggestion. We have added a sentence to emphasize the impacts of our research. 

In-text lines 485-494: This study, the first description of the spatiotemporal distribution of bottlenose dolphins within the Chesapeake Bay, provides a baseline from which future patterns of occurrence can be compared. Additional collection of acoustic and environmental data would provide context to the sightings made by citizen scientists and aid in determining the behaviors, including foraging, of dolphins in the Bay. This behavioral context would improve both the ecological understanding and management of bottlenose dolphins within the Chesapeake Bay. This study is the first description and model of the spatiotemporal distribution of bottlenose dolphins within the highly urbanized Chesapeake Bay. These findings can be used by resource managers to minimize the impacts of the many current and proposed anthropogenic activities in this region.

---

## [Editor Report · Decision Letter 2]

30 Apr 2021

Spatial and temporal variation in the occurrence of bottlenose dolphins in the Chesapeake Bay, USA, using citizen science sighting data

PONE-D-20-27115R2

Dear Dr. Rodríguez,

We’re pleased to inform you that your manuscript has been judged scientifically suitable for publication and will be formally accepted for publication once it meets all outstanding technical requirements.

Kind regards,

Susana Caballero, PhD

Academic Editor

PLOS ONE

Additional Editor Comments (optional):

Thank you for addressing the suggestions and corrections made by the reviewers on your previous version. It is now much clearer.
---

## [Editor Report · Acceptance letter]

6 May 2021

PONE-D-20-27115R2 

Spatial and temporal variation in the occurrence of bottlenose dolphins in the Chesapeake Bay, USA, using citizen science sighting data 

Dear Dr. Rodriguez:

I'm pleased to inform you that your manuscript has been deemed suitable for publication in PLOS ONE. Congratulations! Your manuscript is now with our production department. 

Kind regards, 

on behalf of

Dr. Susana Caballero 

Academic Editor

PLOS ONE